# Self-Stabilizing Capacitated Vertex Cover Algorithms for Internet-of-Things-Enabled Wireless Sensor Networks

**DOI:** 10.3390/s22103774

**Published:** 2022-05-16

**Authors:** Yasin Yigit, Orhan Dagdeviren, Moharram Challenger

**Affiliations:** 1International Computer Institute, Ege University, Bornova, 35100 İzmir, Turkey; yasin.yigit@hotmail.com.tr; 2Department of Computer Science, University of Antwerp, 2020 Antwerpen, Belgium; moharram.challenger@uantwerpen.be; 3AnSyMo/Cosys Core-Lab, Flanders Make Strategic Research Center, 3001 Leuven, Belgium

**Keywords:** wireless sensor networks, internet of things, self-stabilization, capacitated vertex cover, energy efficiency

## Abstract

Wireless sensor networks (WSNs) achieving environmental sensing are fundamental communication layer technologies in the Internet of Things. Battery-powered sensor nodes may face many problems, such as battery drain and software problems. Therefore, the utilization of self-stabilization, which is one of the fault-tolerance techniques, brings the network back to its legitimate state when the topology is changed due to node leaves. In this technique, a scheduler decides on which nodes could execute their rules regarding spatial and temporal properties. A useful graph theoretical structure is the vertex cover that can be utilized in various WSN applications such as routing, clustering, replica placement and link monitoring. A capacitated vertex cover is the generalized version of the problem which restricts the number of edges covered by a vertex by applying a capacity constraint to limit the covered edge count. In this paper, we propose two self-stabilizing capacitated vertex cover algorithms for WSNs. To the best of our knowledge, these algorithms are the first attempts in this manner. The first algorithm is stabilized under an unfair distributed scheduler (that is, the scheduler which does not grant all enabled nodes to make their moves but guarantees the global progress of the system) at most $O(n^{2})$ step, where *n* is the count of nodes. The second algorithm assumes 2-hop (degree 2) knowledge about the network and runs under the unfair scheduler, which subsumes the synchronous and distributed fair scheduler and stabilizes itself after O(n) moves in O(n) step, which is acceptable for most WSN setups. We theoretically analyze the algorithms to provide proof of correctness and their step complexities. Moreover, we provide simulation setups by applying IRIS sensor node parameters and compare our algorithms with their counterparts. The gathered measurements from the simulations revealed that the proposed algorithms are faster than their competitors, use less energy and offer better vertex cover solutions.

## 1. Introduction

Wireless sensor networks (WSNs) do not have a predefined structure to maintain fundamental data-transfer operations. They are crucial communication layer technologies for providing environmental sensing operations in the Internet of Things (IoT). Most of the time, WSNs are deployed for various applications in forests, mines and land borders, where they should bear harsh circumstances [1,2,3]. In such environments, tiny sensor motes can malfunction due to natural challenges. Fault tolerance is an important property to deal with these kinds of challenges. Self-stabilization is one of the best candidates for WSNs to provide fault tolerance and to deal with their ad hoc nature.

A sensor node can leave WSN due to software failures, battery drains and other similar faults. A distributed setting like WSN is considered self-stabilizing if it reaches a legitimate state in case of node leaves. More formally, a system is self-stabilizing if and only if, despite the arbitrary initial state, at least one privileged node will always exist and the system eventually reaches a legitimate state within a limited number of moves [4]. Thus, a self-stabilizing system is expected to reach the correct behavior without any external aid when a fault occurs or starts with an arbitrary configuration. A self-stabilizing algorithm continues to execute even if it does not reach the desired configuration. A self-stabilizing algorithm should provide convergence and closure properties. The convergence property is satisfied if the system reaches the desired state in a finite time. The closure property is preserved if the acceptable final state does not change until the occurrence of any fault.

A self-stabilizing algorithm consists of mutually exclusive rules that are formed as follows: <name>:<precondition>→<action>, where precondition is a boolean predicate which allows nodes to take an action, and action is a series of statements that assign new values to the variables of the nodes and transmit messages if the message passing model is used [5]. In a self-stabilizing algorithm, each node maintains its variables by holding the precondition. If the precondition is true, the node is enabled to make a move. However, this does not mean that the node is allowed to execute their action immediately. The scheduler decides on which nodes could execute their rules according to spatial and temporal properties. Spatial scheduling defines the subset of enabled nodes to be privileged to make moves. While the central scheduler allows only one enabled node in every round, the synchronous scheduler does not restrict the enabled nodes to make their moves in every round. A distributed scheduler allows a subset of enabled nodes in every round. Temporal scheduling refers to the fairness of schedulers in two main types, fair and unfair (i.e., adversarial). An unfair scheduler does not promise all enabled nodes to make their moves but guarantees the global progress of the system [5]. In this paper, we use unfair distributed scheduler which subsumes both central and synchronous schedulers, and it is more realistic for WSN applications.

A WSN can be modeled as a graph *G*(*V*,*E*), where *V* and *E* represent the set of vertices (nodes) and edges (communication links), respectively. A vertex cover of a given undirected graph *G*(*V*,*E*) is a set S⊆V such that each e∈E is incident to at least one vertex of *S*.  Vertex cover is a very useful structure for various WSN applications such as routing, clustering, backbone formation, link monitoring, replica management, network attack protection, etc. [6,7,8,9,10]. Considering the link monitoring application, the number of monitor nodes should be minimized since they should be equipped with extra software/hardware solutions to monitor the network traffic. On the other side, the optimization version of the minimum vertex cover problem, which aims to solve the problem by selecting the minimum number of nodes to cover the whole graph, is in the NP-Hard complexity class [11]. The capacitated vertex cover problem is the generalized version of the classical vertex cover problem that restricts the number of covered edges by $cap_{v}$ for v∈V in the given graph *G*(*V*,*E*) [12]. This restriction property is very useful for WSN applications. For example, limiting the link count monitored by a node directly provides energy efficiency for link monitoring applications. Although there are notable number of VC-related studies, the capacitated version of the problem is rarely investigated in the literature. These capacitated algorithms are central approaches, and they are not distributed and self-stabilizing.

In this paper, we tackle the self-stabilizing capacitated vertex cover problem for WSNs. The contributions of the study are listed as follows:We provide two self-stabilizing capacitated vertex cover algorithms that are based on new heuristics for WSNs. The first proposed algorithm (SS-CVC${}_{1}$) is a modification of Ikeda’s [13] algorithm. The second proposed algorithm, which uses the greedy technique (SS-CVC${}_{2}$), is a novel algorithm for the problem. To the best of our knowledge, this paper is the first attempt to solve distributed self-stabilizing capacitated vertex cover problem for WSNs in the literature.We theoretically analyze our proposed capacitated vertex cover algorithms to ensure their proof of correctness in the self-stabilizing setting. We also analyze the step complexity of our algorithms in order to obtain the efficiency of their resource consumption.We simulate our proposed capacitated vertex cover algorithms and comprehensively evaluate the measured performance results along with the existing self-stabilizing algorithms, where these previous algorithms are modified to provide the capacitated solution. We reveal that our proposed algorithms clearly outperform their counterparts in terms of vertex cover size, the energy consumption (the total energy required for the algorithm execution in sensor nodes) and the time required for stabilization process.

The remainder of this paper is organized as follows. In Section 2, related works on the vertex cover problem are discussed. The formulation of the problem is given in Section 3. After explaining the proposed algorithms in Section 4, we provide the correctness and self-stabilization proof of the algorithms in Section 5. Performance evaluations of algorithms are widely discussed in Section 6. Lastly, we conclude the findings of the study in Section 7.

## 2. Related Work

Various optimization problems in graph theory attract many researchers [14,15]. VC, one of the well-known graph-theoretical optimization problems, is studied for sequential and distributed settings in the literature. The minimum VC problem cannot be approximated within a ratio of 1.36 [16]. The best algorithm which uses matching for the problem was proposed by [17] within a ratio $2-\frac{1}{\log n}$. In the sequential setting, there are a lot of studies that exploit different types of techniques such as depth-first search, local search, dynamic threshold, semi-definite relaxation, graph-theoretic and quantum annealing techniques to provide solutions [18,19,20,21,22,23,24]. The algorithm given in [24] is a metaheuristic-based approach to solve minimum weighted connected VC problem for WSNs. This approach combines a greedy heuristic with a genetic search to decrease the total weight of the solution and the time needed for execution. Since our proposed algorithms are designed for the distributed WSN settings with the qualification of self-stabilizing and capacitated properties, these central algorithms are out of scope for our study.

The graph matching technique is widely exploited to solve the VC problem in distributed settings. Polishchuk has adapted Hancowiak’s distributed graph-matching algorithm to the vertex cover problem with a three-approximation ratio [25,26]. Hoepman’s matching algorithm has been used to solve the vertex cover problem with a two-approximation ratio [27,28]. Since an edge could be covered by its two endpoints, matching-based vertex cover algorithms do not guarantee an approximation ratio lower than 2. Parnas and Ron have provided an algorithm in which each node puts itself in the solution set if its degree δ is greater than $\frac{\Delta }{R}$, where Δ represents the maximum degree of the graph, and *R* is the round count of the algorithm. Kavalci et al. have presented an algorithm that integrates breadth-first search construction process with vertex cover problem for WSNs [28]. Yigit et al. have proposed two novel algorithms for WSNs, which integrate breadth-first search similar to Kavalci’s algorithm [10]. Along with the distributed setting, many studies have been conducted in parallel settings with different techniques such as membrane computing [29], digital annealing [30] and massively parallel computing [31].  To solve matching problem for WSNs, please refer to the algorithms [32,33,34]. Since these matching-based algorithms do not provide a capacitated and self-stabilizing solution, we exclude these algorithms in our study.

The capacitated vertex cover problem is generally studied in terms of linear programming and relaxation. The capacitated vertex cover problem was introduced in [12] for weighted graphs. Guha proposed two algorithms for the problem in the central setting. The first algorithm is a relaxation which solves a linear formulation of the capacitated vertex cover within a 4-approximation ratio. The second is the dual of the first and has an approximation ratio of 2. Chuzhoy and Naor have proposed two randomized algorithms to solve the capacitated vertex cover, which provide 8- and 3-approximation ratios with linear a programming formulation [35]. Gandhi et al. have introduced a 2-approximation algorithm which consists of a pre-processing step to reduce the vertex cover size [36]. In this pre-processing step, each capacity-1 vertex is excluded from the solution set until the solution does not satisfy feasibility, which is checked by using a max-flow algorithm. For a recent study concerning the performance of the capacitated algorithms in WSNs, please refer to [37]. All of the capacitated algorithms are not distributed; hence, they cannot provide a self-stabilizing solution for vertex cover.

Kiniwa has proposed the first self-stabilizing vertex cover algorithm which exploits the matching technique to obtain the vertex cover [38]. Kiniwa’s algorithm firstly constructs a maximal matching by favoring the edges connecting the heaviest nodes with the lightest nodes on the graph and then covers the nodes on the basis of this matching. Each vertex holds cover and color variables, which describe whether the vertex is in a vertex cover set and the color of the matched port, respectively. Additionally, each vertex maintains three sets, which are named High, Low and Others. The High(v) set contains neighbors of vertex *v* that have a larger color value. On the contrary, Low(v) contains neighbors of node *v* which have a smaller color value. Others(v) holds neighbors that do not point to *v*. The algorithm obtains a $\left(2-\frac{1}{\Delta }\right)$-approximation vertex cover using shared memory and distributed scheduler in the M+2 round, where *M* is the size of the matching.

In [39], Turau et al. have introduced two self-stabilizing vertex cover algorithms, which run on anonymous networks. Both algorithms are based on the study given in [25] which cannot exceed the 3-approximation ratio by using the matching method on the anonymous networks proved by [40]. The first algorithm consists of two predicates. The basic algorithm calculates the 3-approximation ratio vertex cover set with O(n+m) moves, where *n* and *m* are the number of vertices and edges, respectively. In the same article, Turau presents an improvement method that approximates the optimal vertex cover solution up to $\left(3-\frac{2}{\Delta +1}\right)$ times in O(n+m) moves. This improvement algorithm adds new rules to the first basic algorithm. After the execution of two basic rules, nodes that are still not matched with both pointers are excluded from the solution set.

Many self-stabilizing maximal independent set algorithms have been proposed by the researchers over the years [13,41,42,43,44]. The vertex cover and the independent set solutions complement each other on a given graph *G*(*V*,*E*). Based on this fact, the performance evaluation of the vertex cover and independent set algorithms for WSNs was examined in [45] in the self-stabilizing setting. These algorithms provide self-stabilization solutions but do not give capacitated solutions. We modified these self-stabilizing vertex cover algorithms so as to provide capacitated solutions in this study, and the simulation results show that our proposed algorithms outperformed their counterparts (modified algorithms in the literature) in terms of vertex cover solution and time consumption.

## 3. Problem Formulation

An example sensor network deployment for a habitat monitoring application is depicted in Figure 1a, where there are 12 nodes in the sensing area and node 1 is the sink node. The graph representation of this network is given in Figure 1b. In Figure 1c, link monitoring application for this topology is shown. In this application, each link must be sniffed by one secure point (monitor node) to detect attacks such as packet injection and data manipulation. The red nodes (nodes 1, 3, 4, 5, 6 and 8) are secure points that are assigned to control message traffic in Figure 1c. Red arrows show the assigned links to the monitor nodes in the same figure. For example, the links (8,9) and (8,10) are monitored by node 8. This architecture can also be used in other common operations such as backbone formation, clustering and routing. Red nodes can be cluster heads, and ordinary nodes can send their data to the cluster heads to achieve data aggregation. The network induced by red nodes is a virtual backbone that can carry messages to the sink node. By accomplishing the clustering and backbone formation operations, the data packets can be routed from ordinary nodes to the sink node.

There are two versions of the capacitated vertex cover problem, which are named soft and hard capacitated. While the nodes cannot exist more than once in the solution set in the hard capacitated version, there is no restriction about the existence count of vertices for the soft capacitated vertex cover problem. In this paper, we tackle with the soft capacitated version of the problem. If a link monitoring application uses soft capacitated vertex cover, then a node may create more than one process, where each of these processes are assigned to monitor the links. In other words, the existence count of a vertex is equal to the process count created on a sensor node in the soft capacitated version. We recognize the following assumptions regarding the network:Each node is represented by a distinct identifier (i.e., id).The communication channels between nodes function in both directions.Nodes do not include any GPS-based position tracking; hence, they are unaware of their positions in the network.All nodes are equipped with the same hardware and run the same algorithm.For the SS-CVC${}_{1}$ algorithm, each node knows its 1-hop (degree-1) neighbors that are one hop away from the node and directly communicates with them.For the SS-CVC${}_{2}$, we assume that each node knows 2-hop (degree-2) neighbors and can send messages to them through 1-hop neighbors.

In this study, our objective is to develop heuristic-based self-stabilizing capacitated vertex cover algorithms (SS-CVC) for WSNs that are efficient in terms of energy consumption, time usage and the size of the vertex cover set. Therefore, we make a list of our goals as follows:To fulfill the convergence property, the algorithms must reach the stable configuration (in which nodes do not make a move) as fast as they can within a minimum number of moves and steps. Hence, wallclock execution times of algorithms (the time passed during the execution) should be low. In a stable configuration, <precondition> predicates of all nodes are false when the rules are defined in <name>:<precondition>→<action> form.Having become stabilized, the network should preserve capacitated vertex cover until a fault happens in order to satisfy the closure property.The algorithms should be energy-efficient for the sake of increasing the lifetime of the network. The energy consumption of an algorithm is the total energy required for its execution in sensor nodes.When a failure occurs, such as the shut down of a node, each active node should be able to initiate the self-stabilization procedure itself.Since WSNs are deployed over harsh areas, some nodes can leave and new nodes can join the network. In such cases, the self-stabilization process should occur without any external intervention.The VC construction process should be independent of the location information of the nodes.

## 4. Proposed Algorithms

In this section, the proposed algorithms are explained and exemplified in the sample networks. We firstly introduce the maximal independent set based algorithm, which is called SS-CVC${}_{1}$. After the first algorithm, we introduce our second algorithm, which needs 2-hop information about the network.

### 4.1. Ss-Cvc${}_{1}$ Algorithm

In this subsection, we present the SS-CVC${}_{1}$ algorithm, which is based on Ikeda’s MIS algorithm [13]. Ikeda’s MIS algorithm runs under the unfair scheduler and stabilizes itself at most $O(n^{2})$ steps, where *n* is the node count. We modified the first two rules of Ikeda’s algorithm to obtain a vertex cover set. Each node i∈V maintains $covered_{i}$ variable (the output of the algorithm) that has two different states: {0,1}. After the first two rules, we add two additional rules that satisfy the capacity constraint. Algorithm 1 shows the proposed SS-CVC${}_{1}$ algorithm. The rules are mutually exclusive, and since an unfair distributed scheduler is used, any enabled node can make a move. If there are no enabled nodes (meaning that any rule of a node is not true), there will be no move.
**Algorithm 1:** SS-CVC${}_{1}$**process***i*N(i) neighbors of node *i***Variables:**$\;covered_{i}\in \left\{0,1\right\}$$\;nex_{i}\in N^{+}$**Rules:****R1:**$covered_{i}=1\wedge \forall j\in N\left(i\right):covered_{j}=1$$\;\Rightarrow covered_{i}:=0$**R2:**$covered_{i}=0\wedge \exists j\in N\left(i\right):covered_{j}=0\wedge cost_{1}\left(j\right)>cost_{1}\left(i\right)$$\;\Rightarrow covered_{i}:=1$**R3:**$covered_{i}=1\wedge nex_{i}\neq \left\lceil \frac{\left|N(i)\right|}{cap_{i}}\right\rceil$$\;\Rightarrow nex_{i}:=\left\lceil \frac{\left|N(i)\right|}{cap_{i}}\right\rceil$**R4:**$covered_{i}=0\wedge nex_{i}\neq 0$$\;\Rightarrow nex_{i}:=0$

**R1** is a simple rule that node *i* changes its $covered_{i}$ variable by checking its neighbors *j*’s $covered_{j}$ variable. If the $covered_{i}$ variable of the node *i* is 1 and all of its neighbors *j*’s $covered_{j}$ variables are 1, node *i* changes its $covered_{i}$ variable to 0. Figure 2a shows a sample scenario from *i*’s point of view, where all neighbors of *i* are already covered, thus *i* sets $covered_{i}$ as 0.

In **R2**, each node decides whether or not to join the vertex cover set by looking at the neighbors’ $covered_{j}$ and $cost_{1}\left(j\right)$. A vertex *i* simply calculates its cost(i) with the formula given in Equation (). The algorithm chooses the vertex with the local minimum cost. In Figure 2b, the node *i* enables **R2** since its cost is lower than *v*:   
(1)$$cost_{1}\left(i\right)=\frac{\left\lceil \frac{\left|N(i)\right|}{cap_{i}}\right\rceil *weight_{i}}{\left|N(i)\right|}$$

**R3** and **R4** regulate the $nex_{i}$ variable that is the number of existence in vertex cover set for vertex *i*. If $covered_{i}$ is 1 and $nex_{i}$ is not correct for vertex *i*, as seen in Figure 2c, the vertex sets $nex_{i}$ to $\left\lceil \frac{N(i)}{cap_{i}}\right\rceil$. On the other hand, if $covered_{i}$ is 0 and $nex_{i}$ is not equal to 0, the vertex sets it to 0, as illustrated in Figure 2d.

In Figure 3, an example operation for the proposed SS-CVC${}_{1}$ algorithm is shown. The algorithm starts with an arbitrary initial configuration where each vertex is labeled as (cap, weight, nex). Red color represents that the vertex is already in the vertex cover set. At the given initial configuration, the costs of vertices are 5, 2, 2, 3 and 2, respectively. Although vertex 1 and vertex 2 have the same cost of 2, the algorithm uses the vertex identifier to prevent the neighbor nodes from entering into the vertex cover set together. In the first round, vertices 2 and 4 execute **R2** to set their covered variable to 1. Vertex 1 executes **R4** and sets $nex_{1}$ to 0. Vertex 3 executes **R3** to justify $nex_{3}$ to 2. In the second round, vertex 3 executes **R1** and sets $covered_{3}$ variable to 0 since all of its neighbors are currently in the vertex cover set. In addition, vertex 4 sets $nex_{4}$ to 2 by executing **R3**. In the last round, vertex 3 executes **R4** to set $nex_{3}$ to 0. SS-CVC${}_{1}$ produces the optimal solution, whose total weight is 9 for this example.

### 4.2. SS-CVC${}_{2}$ Algorithm

In this subsection, we improve our SS-CVC${}_{1}$ algorithm to reduce the weight of the produced vertex cover solution and the round complexity. In order to achieve this, we facilitate a model defined by Turau in [46]. The model is named as the expression model, in which each vertex *i* has expressions that show their own states and states of their neighbors. The 2-distance model is a special case of the expression model. In the expression model, each vertex not only reads the states of its neighbors but also reads the expressions of all of its neighbors in an atomic step. We present the SS-CVC${}_{2}$ capacitated self-stabilizing vertex cover algorithm consisting of two expressions. The pseudo-code of the proposed algorithm is given in Algorithm 2. The output of the algorithm is $covered_{i}$ for node *i*.

The expression is_tight(i) checks if the node *i* is already in vertex cover set and $nex_{i}$ is correct. We use this expression to include a vertex in the vertex cover. The expression is_candidate(i) returns true if a node *i* has a locally optimal cost among all of its neighbors N(i) which can enter the vertex cover set. The macro $trade\_off_{i}$ is used to calculate the trade-off value of a vertex. We need to calculate trade-off value for vertex *i*, so we count the double-covered edges and then multiply this value with the payback value $\frac{nex_{i}\times weight_{i}}{covered\_by\_me_{i}}$ for an edge.

The expression $has\_max\_trade\_off_{i}$ is used to find the locally optimal vertex to exclude it from the vertex cover set. If a node has a max trade-off among all of its neighbors with double covered edges and is already tight, this expression returns true. Note that the priority of each rule is defined by its sequence number.

**R1** selects a locally optimal node to enter the solution set by calculating the cost of each node that is candidate to enter this set. To calculate the cost of a vertex *i*, Equation () is used. If node *i* activates **R1** and is privileged by the scheduler, it sets $covered_{i}$ to 1, $nex_{i}$ to the correct value and covers all edges in $uncovered\_edges_{i}$. Figure 4a gives an example configuration for activation of **R1** from *i*’s perspective. In this configuration, *i* is not tight because $covered_{i}=0$ and covers edge (i,k) but edges (i,j) and (i,l) are not covered by others. The cost of *i* is lower than its neighbors for this scenario; therefore, the only enabled node becomes *i*, which enables **R1**: (2)$$cost_{2}\left(i\right)=\frac{\left\lceil \frac{\left|uncovered\_edge\right|}{capacity_{i}}\right\rceil *weight_{i}}{\left|uncovered\_edge\right|}$$
**Algorithm 2:** SS-CVC${}_{2}$.
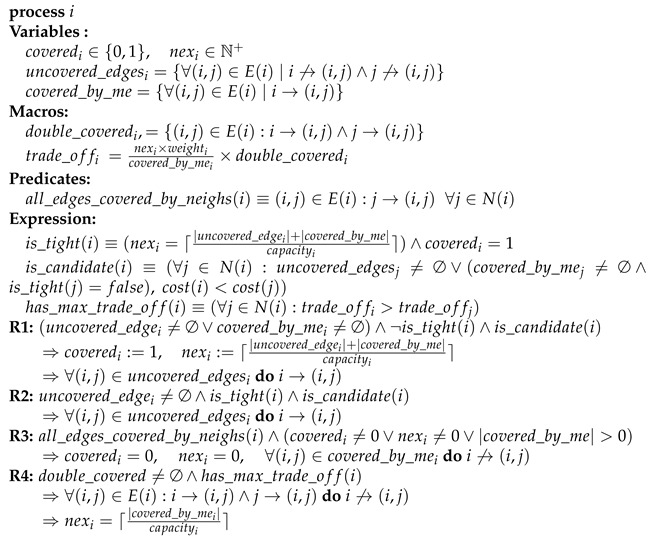


If node *i*’s $uncovered\_edge_{i}$ set is not empty but the node is tight, the node checks whether it is a candidate. If the node is a candidate to join the vertex cover set, it enables **R2**. In Figure 4b, node *i* activates **R2** since it is a locally optimal candidate to join the vertex cover set, and it already satisfies the tightness that distinguishes **R1** and **R2**. If the node makes a move, it just covers all edges in uncovered_edge_i so $covered_{i}$ and $nex_{i}$ variables become correct. In **R3**, if at least one of $covered_{i}$, $nex_{i}$, |covered_by_me| variables of a node *i*, which means that all its incident edges are covered by all of its neighbors, are not 0, the node executes **R3** and resets these variables. As seen in Figure 4c, node *i* activates **R3** as all its neighbors cover all incident edges to it but $covered_{i}=0$. We use **R4** to exclude unnecessarily selected vertices from the solution set by calculating their trade-off value. As depicted in Figure 4d, the edge (i,j) is covered by two endpoints, each node calculates its trade-off value, and the node with the maximum trade-off value, *i* in this case, reduces its nex variable. If the newly calculated nex variable is 0, the node sets covered variable to 0 as well.

Figure 5 shows the execution of the SS-CVC${}_{2}$ algorithm on the arbitrary initialized network. Nodes are labeled as in the execution of SS-CVC${}_{1}$ in Figure 3. We also randomly initialize covered_by_me sets for each vertex. The red vertices represent the covered vertices, and the red arrows show the edges covered by the vertices. In Round 1, vertex 2 excludes itself from the vertex cover set by executing **R4** since its $trade\_off_{2}$ value 8 is greater than vertex 3. Vertex 4 excludes itself from the vertex cover set by executing **R3** since all of its edges are covered by its neighbors. Vertex 3 covers itself by running **R1** with $cost_{1}=0.75$ as the local minimum cost.

In Round 2, vertices 1 and 6 execute **R1** since they have 3 and 2 costs, respectively, which are the local minimum. Moreover, vertices 0 and 5 execute **R1** since they have 1 cost in Round 3. Each vertex contributes to the solution set only once; therefore, the total weight of this solution is 15, that is, the optimal solution. Although we find an optimal solution for this example, we do not grant an optimal solution for all networks since our algorithms are heuristic-based approaches for WSNs.

## 5. Theoretical Analysis of Algorithms

In this section, proof of the correctness and step complexities of the algorithms are provided.

### 5.1. Theoretical Analysis of SS-CVC${}_{1}$

**Theorem** **1.**
*The SS-CVC${}_{1}$ algorithm stabilizes after $O(n^{2})$ steps under an unfair distributed scheduler.*


**Proof.** Assume that the scheduler immediately gives permission to all nodes that want to execute **R1** and **R3** at the same time and gives permission one by one to the other nodes that want to execute **R2** and **R4**. Consider a configuration as shown in Figure 6, whose cost values increase in the following order: cost(0)<cost(1)<…<cost(n−1).The scheduler allows all nodes to make their moves in one step for the first phase. In the second phase, all nodes are privileged one by one, and this process takes *n* steps. The third phase is ended in only one step because each node wants to execute **R1** or **R3**. In phase 4, nodes n−1 and n−2 stabilize and do not want to make any move. Thus, the rest of the nodes make their moves in n−2 steps. The count of nodes which want to make a move decreases by two in each consecutive two rounds. We can formalize this relation as 1+n−(2×r), where *r* is the sequence number of two consecutive phases. Such a scenario is shown in Figure 6, in which the system stabilizes when $r=\frac{n-1}{2}$. We can formulate as follows:   
(3)$$\begin{array}{cc} \; & \sum_{r=0}^{\frac{n-1}{2}}1+n-\left(2\times r\right)=\sum_{r=0}^{\frac{n-1}{2}}1+\sum_{r=0}^{\frac{n-1}{2}}n-\sum_{r=0}^{\frac{n-1}{2}}2r\\ \; & =\frac{\left(n+1)(2+2n-n+1\right)}{4}=\frac{\left(n+1)(n+3\right)}{4}\\ \; & =O(n^{2}) \end{array}$$When we solve the summation formula, we reach the $O(n^{2})$ step, and this concludes the proof of the theorem.    □

### 5.2. Theoretical Analysis of SS-CVC${}_{2}$

In this subsection, we prove that SS-CVC${}_{2}$ is a self-stabilizing capacitated vertex cover algorithm. Firstly, we show the SS-CVC${}_{2}$ algorithm producing a capacitated vertex cover solution when it reaches the stable configuration. Afterwards, we will show that the algorithm reaches stable configuration in a finite number of moves under the unfair distributed scheduler.

**Lemma** **1.**
*When the algorithm is in stable configuration, $\forall e\in \left(i,j\right)\in E:\left(i\to \wedge \;covered_{i}=1\right)\vee \left(j\to e\wedge covered_{j}=1\right)$.*


**Proof.** If there is such an edge as e=(i,j), both endpoints i,j do not cover *e*, *i* or *j* executes **R1** or **R2** and sets covered=1.    □

**Lemma** **2.**
*In the stable configuration, $\forall i\in V:nex_{i}=\left\lceil \frac{covered\_by\_me_{i}}{capacity_{i}}\right\rceil$.*


 **Proof.** If $nex_{i}$ variable of a vertex *i* is not equal to $\left\lceil \frac{covered\_by\_me_{i}}{capacity_{i}}\right\rceil$, the vertex updates its $nex_{i}$ according to the following four methods:
If is_tight(i) expression of *i* is false and neither covered_by_me nor uncovered_edges are empty, *i* executes **R1** and updates $nex_{i}=\left\lceil \frac{|uncovered\_edge_{i}\left|+|covered\_by\_me\right|}{capacity_{i}}\right\rceil$. After the execution of **R1**, all edges of $uncovered\_edges_{i}$ are covered by *i*.If vertex *i* is tight but it has at least one edge in $uncovered\_edges_{i}$, it executes **R2** to sets $nex_{i}=\left\lceil \frac{|uncovered\_edge_{i}\left|+|covered\_by\_me\right|}{capacity_{i}}\right\rceil$ and covers all edges in $uncovered\_edges_{i}$.If all incident edges to *i* are covered by N(i), *i* executes **R3** and sets $nex_{i}=0$ and uncovers all edges that are covered by itself.If an edge e=(i,j) is covered by both endpoints, the edge is considered as a double-covered edge. The vertex *i* that has the maximum trade_off among its one-hop local neighborhood, uncovers the double covered edges and updates its $nex_{i}$ by $covered\_by\_me_{i}$.
   □

**Lemma** **3.**
*In the stable configuration, an edge e=(i,j) is covered by its one endpoint i or j.*


**Proof.** Assume a situation in which *e* is covered by both endpoints. The nodes *i* or *j* will execute **R4** according to their trade_off(). This is a contradiction and proves the lemma.    □

**Theorem** **2.**
*A stable configuration SS-CVC${}_{2}$ is a capacitated vertex cover in which the size of solution set cannot be decreased by removing a vertex.*


**Proof.** When the system stabilizes, the variable covered of an endpoint of an edge is 1 (Lemma 1). According to Lemma 2, nex variables of the covered vertices are equal to $\left\lceil \frac{covered\_by\_me_{i}}{capacity_{i}}\right\rceil$. If a vertex provides these properties, the vertex is considered tight.If all neighbors of vertex *i* already cover each connected edge, the node executes **R3**. To prevent double-covered edges, **R4** is executed by vertices which have the locally maximal trade-off.    □

**Lemma** **4.**
*Each vertex could execute either **R1** or **R2** only once.*


**Proof.** **R1** and **R2** are used to enter the vertex cover set and are mutually exclusive rules due to is_tight() expression. Once a vertex enters the vertex cover set, it executes neither **R1** nor **R2** until a fault occurs.    □

**Lemma** **5.**
*Each vertex executes **R3** only once.*


**Proof.** Due to the arbitrary initial configuration property of self-stabilization, all edges are covered by all vertices in the initial configuration. In such a configuration, all nodes except the one which executes **R1** or **R2** execute **R3** only once. In the configuration, where a vertex *i* has vertices in $covered\_by\_me_{i}$ but its cost is greater than its neighbor’s cost, *i* executes **R3** only once when its neighbors enter the solution set.    □

**Lemma** **6.**
*Each vertex executes **R4** only once.*


**Proof.** Two neighbor vertices could cover the same edge due to the arbitrary initial configuration or execution of the algorithm. To prevent this, each node with double-covered edges and maximum trade-off value executes **R4** only once during the execution of the algorithm.    □

**Lemma** **7.***If a vertex executes****R3****, it does not execute****R4***.


**Proof.** If a vertex *i* makes a **R3** move, then it does not have double-covered edges due to the removal of the edges of $covered\_by\_me_{i}$.    □

**Theorem** **3.**
*The step complexity of the SS-CVC${}_{2}$ heuristic algorithm under the unfair distributed scheduler is O(n).*


**Proof.** The unfair distributed scheduler does not guarantee the privilege to activate all nodes in any round but at least one node in one round. Due to the Lemmas 4 and 7, each vertex can make a maximum of two moves. According to the definition of the unfair scheduler, it takes 2*n* steps to stabilize the system. According to this, the step complexity of SS-CVC${}_{2}$ is O(n).    □

## 6. Performance Evaluation

### 6.1. Experimental Setup

We implement algorithms on the self-stabilizing simulator proposed in [47]. The simulator has been written with the widely used programming language Python. The simulator provides three main types of schedulers with fair and unfair options. Each node is represented as a class object and holds its variables as attributes. Having made a move, the node changes its variables and sends the new set of variables to all of its neighbors. Each node tracks the sent and received bytes which vary for each algorithm according to the size of the messages (the total size of the fields in the messages). The simulator also provides an infrastructure to make it possible to implement algorithms that need 2-hop information about topology. We run the algorithms under the unfair distributed scheduler, which is the most restricted scheduler type, since it does not guarantee that all active nodes are privileged eventually. The scheduler prevents nodes from making moves with 0.5 probability, but guarantees global progress by privileging at least one vertex in each step.

We used random geometric network models in order to simulate WSNs. Each vertex of the given graph *G(V,E)* is scattered to a 2D area *A*. The sizes of the randomly generated WSNs vary from 50 to 300 (with 50 steps), in which the nodes are considered connected if the Euclidean distance between two of them is smaller than their transmission range *r*, as applied in [47]. WSNs are classified for various densities that have three, five, seven and nine average degrees. Each performance metric is obtained with 30 different simulation scenarios. We randomly assign a weight to each vertex in the interval [1−50]. Furthermore, we assign to each vertex *i* a cap value in the interval [1−Δ], where Δ represents the maximum degree of a network.

In addition to the algorithms we proposed, we implemented the algorithms of Kiniwa [38] and Turau [39], which are state-of-the-art algorithms for self-stabilizing vertex cover domain. To provide the capacity constraint for the algorithms of Kiniwa and Turau, we added nex and cap and weight variables and two rules which are shown in Algorithm 3. Furthermore, we provided the 1-hop implementation of the SS-CVC${}_{2}$ algorithm thanks to the transformer proposed in [46].
**Algorithm 3:** Additional rules.**process***i***R1:**$covered_{i}=1\wedge nex_{i}\neq \left\lceil \frac{\left|N(i)\right|}{cap_{i}}\right\rceil$$\;\Rightarrow nex_{i}:=\left\lceil \frac{\left|N(i)\right|}{cap_{i}}\right\rceil$**R2:**$covered_{i}=0\wedge nex_{i}\neq 0$$\;\Rightarrow nex_{i}:=0$

All variables for each node are initialized randomly before the algorithm starts. When a node changes its state, all 1-hop neighbors can see this move (For SS-CVC${}_{2}$, 2-hop information is provided). We compared the algorithms in terms of move count, step count, the total weight of vertex cover, and cardinality of vertex cover multi-set. In addition, we measured important WSN metrics such as sent bytes, received bytes and energy consumption. The energy consumption of an algorithm is the energy required for its execution. Move count plays a crucial role in the WSN because a node must send their new state to its neighbor after a move. The count of moves has a direct impact on the complexity of the message. Step complexity is important to see how long it takes to reach the stable algorithm configuration (the time passed during the execution of the algorithm in WSN). The weight and cardinality of the vertex cover are other important metrics to facilitate when comparing the algorithms, since we want to formulate a desirable solution in the shortest possible time. Although Kinawa’s and Turau’s algorithms have been proposed for unweighted graphs, we carried out weighted experiments to compare all algorithms.

Thereafter, Turau’s basic and improved algorithms and Kiniwa’s algorithm will be called TURAU${}_{1}$, TURAU${}_{2}$ and KINIWA, respectively. The transformed version of the SS-CVC${}_{2}$ algorithm is called T-SS-CVC${}_{2}$.

### 6.2. Evaluations

Move counts of the algorithms are shown in Figure 7 with a fixed average degree and a fixed network size. It is seen clearly that the move counts increase with the number of nodes in the network for each algorithm, as shown in Figure 7a. The KINIWA algorithm makes the maximum number of moves to reach a stable configuration and is followed by TURAU${}_{2}$. In networks with 250 vertices, KINIWA took 1347 moves to stabilize, while our proposed algorithms, SS-CVC${}_{1}$ and SS-CVC${}_{2}$, needed 332 and 309 moves, respectively, on the same networks. The closest algorithm to our algorithm regarding move count is TURAU${}_{1}$, which makes 833 moves until reaching the stable configuration on the 250-sized networks. The density of the networks does not significantly affect the move counts of the algorithms, as seen in Figure 7b. Especially, SS-CVC${}_{1}$ and SS-CVC${}_{2}$ have been minimally affected by density. The SS-CVC${}_{2}$ algorithm makes 173, 184, 192 and 194 moves in networks which have 3, 5, 7 and 9 average degrees, respectively. KINIWA needs 669, 783, 854 and 919 moves on average in the same types of networks. Our proposed algorithm, SS-CVC${}_{2}$, showed 2.5 times better performance than its nearest counterpart, TURAU${}_{1}$, in terms of move counts.

The step count of SS-CVC${}_{2}$ varies between 33 and 68, while SS-CVC${}_{1}$ has less than 34 step counts, as shown in Figure 8a. KINIWA needs 78–148 steps to stabilize for each size of the graph. The algorithm that has the closest step count to our algorithm is TURAU${}_{1}$ with 45 to 74 steps. Compared with TURAU${}_{1}$, the SS-CVC${}_{1}$ and SS-CVC${}_{2}$ algorithms need 1.10 and 2.15 times less steps, respectively, to stabilize. SS-CVC${}_{1}$ and SS-CVC${}_{2}$ are 4.22 and 2.17 times faster than KINIWA, respectively, which has the greatest step size for all graph sizes. As seen in Figure 8b, the step count of SS-CVC${}_{1}$ stayed stable at around 28 as the density of the graph increased, while SS-CVC${}_{2}$’s step count increased with the density. Starting with the average degree of 7, the SS-CVC${}_{2}$ algorithm exceeds the TURAU${}_{1}$ algorithm. However, in most graph types, the SS-CVC${}_{1}$ and SS-CVC${}_{2}$ algorithms outperformed other algorithms in terms of step count.

Figure 9 depicts the sent bytes for each algorithm on the whole network in kB. The SS-CVC${}_{1}$ and SS-CVC${}_{2}$ algorithms need the lowest message passing traffic related to their move count performance, which directly impacts message traffic because nodes must inform their neighbors after each move. Note that the other factor for sent byte performance is the message size. For example, the message size for the SS-CVC${}_{1}$ algorithm is 5 bytes, while SS-CVC${}_{2}$ holds 9 bytes for each package. Due to this difference, the slope of the sent bytes and the move counts line of SS-CVC${}_{2}$ remain stable, but the sent bytes exceeded SS-CVC${}_{1}$, as seen in Figure 9a,b. The size of the network directly influences the number of bytes sent by nodes since a larger network needs more move and message passing. For the networks with 250 nodes, TURAU${}_{1}$, TURAU${}_{2}$ and KINIWA send 3.26 kB, 6.1 kB and 6.57 kB of messages in total, respectively, while the SS-CVC${}_{1}$ and SS-CVC${}_{2}$ algorithms need 1.62 kB and 2.72 kB messages, respectively, to stabilize. The SS-CVC${}_{1}$ algorithm has a two times better performance against its closest competitor, TURAU${}_{1}$. Network density did not play a crucial role on the sent byte performance of algorithms, as seen in Figure 9b.

The received bytes is another important measurement to determine the quality of the algorithm because they affect the lifetime of the network. Figure 10 shows the performances of the algorithms in terms of received bytes for the whole network until the system reaches a stable configuration. Figure 10a compares the received bytes performance of the algorithms with respect to the size of the network. As seen in Figure 10a, there is a linear correlation between received bytes and the size of the network because the increasing node count directly affects the transmitted byte count. The SS-CVC${}_{1}$ algorithm outperformed all other algorithms since it has smaller packages to send and less message traffic. For example, the SS-CVC${}_{1}$ algorithm needed 9.72 kB average received bytes for networks with 300 nodes; on the other hand, KINIWA needed 39.1 kB to stabilize in same type of networks. The best algorithm after SS-CVC${}_{1}$ is the TURAU${}_{1}$ algorithm, which needs two times more received bytes in total. The SS-CVC${}_{2}$ algorithm showed poor results since it assumes messages are passed to the 2-hop neighborhood. Figure 10b shows the performance of the algorithms as the average degree of the networks increases. Increasing the density of the network increases the received byte counter linearly since the transmitted messages reach more neighbors. Unlike the sent byte performance of algorithms, the density of the networks impacts the received byte performance of the algorithms. However, this impact is minimal for SS-CVC${}_{1}$ in comparison to the other algorithms, as seen from the slopes of lines. In networks with 150 nodes, SS-CVC${}_{1}$ always stayed below 9 kB for all density types.
(4)E≈((S×17+R×16)/31.25)×3.3mJ

Figure 11 illustrates the energy consumption of each node in the network. The energy consumption is formulated according to the IRIS datasheet, where each IRIS device consumes 17 mA in transmit mode and 16 mA in receive mode when it works with the maximum transmission power. The devices needs 3.3 V to operate. The transmit data rate of each node is 250 kb/s, which is equal to 31.25 kB/s. By using the energy formula E=V×I×T, we can obtain Equation (4) [48,49]. As seen in Figure 11a, the SS-CVC${}_{1}$ algorithm consumes 67.44 mJ per node as the most energy-efficient algorithm among all implemented algorithms. Since the energy calculation takes into account the received bytes count, the SS-CVC${}_{2}$ algorithm consumes more energy than its counterparts. After the SS-CVC${}_{1}$ algorithm, TURAU${}_{1}$ consumes 136.08 mJ per node to reach the stable configuration. The SS-CVC${}_{1}$ algorithm solves the capacitated vertex cover problem two times more efficiently than its nearest counterpart. Figure 11b shows the impacts of the network density on energy consumption for algorithms. Again, SS-CVC${}_{1}$ has the best performance among the other implemented algorithms.

Figure 12a,b show that the cardinality of the VC solution is directly affected by the size and density of the network. Intuitively, when the network becomes larger and denser, it must choose more vertices to cover all edges in the network because the edge count increases by the node count and the average degree of the network. The SS-CVC${}_{2}$ algorithm produced the best VC solution for all network sizes and densities. The SS-CVC${}_{2}$ produced a VC multiset that contains 291 vertices on the 250-sized network. The SS-CVC${}_{1}$ algorithm is the runner-up with a multiset that contains 360 vertices on the networks that have 250 vertices. The SS-CVC${}_{2}$ algorithm produced 1.75, 1.59 and 1.48 times smaller vertex cover solutions in comparison to TURAU${}_{1}$, TURAU${}_{2}$ and KINIWA, respectively. As the network becomes denser, the SS-CVC${}_{2}$ algorithm is less affected by the density when we compare it against the other algorithms. On the 150-sized networks, SS-CVC${}_{2}$ produced VCs in sizes of 140, 168, 183 and 194 in networks whose average degrees vary between 3 and 9, while the best matching-based algorithm KINIWA produced 183-, 249-, 281- and 320-sized VC solutions.

When we investigate Figure 13, it can be stated that the weight of the VC has the same characteristic as the cardinality of the VC. The SS-CVC${}_{1}$ and SS-CVC${}_{2}$ algorithms produce vertex cover solutions that have less weight than the other algorithms. SS-CVC${}_{1}$, KINIWA, TURAU${}_{2}$ and TURAU${}_{1}$ produced weighted vertex covers which are 1.23, 1.58, 1.70 and 1.88 times bigger, respectively, than SS-CVC${}_{2}$’s solutions for networks in size 300. The density of the networks affects the weight of the solution, as seen in Figure 13 for all algorithms; however, our proposed algorithm SS-CVC${}_{2}$ produced the lowest weights among all other algorithms for all network densities.

We elaborate the approximation ratios of the weights of the algorithms in Table 1, Table 2 and Table 3. We obtained the optimal solution using the SageMath programming language by implementing Guha’s integer linear programming algorithm proposed for capacitated vertex cover in [12]. The implemented algorithm is executed on a server that has an Intel-Xeon E5-2620 v4 processor. The tables show us that the SS-CVC${}_{2}$ algorithm has better approximation ratios among all algorithms. After the SS-CVC${}_{2}$ algorithm, the second lowest approximation belongs to SS-CVC${}_{1}$ for all network types. Matching-based vertex cover algorithms produced more than two approximation ratios. Since these algorithms have not been proposed for weighted networks, they exceeded theoretical approximation ratios. The density of the networks affects the approximation ratios of all algorithms, while the network size has no correlation with the approximation ratio.

We provide the move count and step count performances of the transformer proposed by Turau in Figure 14. The transformer provides an interface that enables executing an algorithm which is designed for 2-hop with 1-hop information with O(m) slow-down factor. The move count and step count of the transformed SS-CVC${}_{2}$ (namely, T-SS-CVC${}_{2}$) are always higher than SS-CVC${}_{2}$ due to slow-down factor. T-SS-CVC${}_{2}$ needs to make moves at least 15.4 times more than SS-CVC${}_{2}$ to reach the capacitated vertex cover solution since the SS-CVC${}_{2}$ algorithm stabilizes after 309 moves while the T-SS-CVC${}_{2}$ algorithm needs 4773 moves to reach the stable configuration on the 250-sized network. We could infer the same with respect to the step count, which is 15 times more for the T-SS-CVC${}_{2}$ on the 50-sized network. To reach a stable configuration, T-SS-CVC${}_{2}$ needs more message traffic, as shown in Figure 15. As mentioned before, message sending is tightly related to the move count of the algorithm. Nodes in T-SS-CVC${}_{2}$ send 15 times more bytes and receive 4 times more message bytes in comparison with SS-CVC${}_{2}$. As shown in Figure 16a, T-SS-CVC${}_{2}$ needs approximately five times more energy to produce a vertex cover set on average. In terms of the weight of the vertex cover, the SS-CVC${}_{2}$ and T-SS-CVC${}_{2}$ algorithms produced the same results as those given in Figure 16b.

## 7. Conclusions

In this study, we proposed two self-stabilizing capacitated vertex cover algorithms for IoT-enabled WSNs. First of all, we modified Ikeda’s algorithm by adding two new rules and changing the existing rules. Furthermore, we proposed a new algorithm based on greedy heuristic for WSNs. The first algorithm, named SS-CVC${}_{1}$, needs 1-hop information about the topology to reach a stable configuration to satisfy the capacitated vertex cover property. SS-CVC${}_{2}$ requires 2-hop information to achieve the goal of capacitated vertex cover. We provided sample executions of both algorithms and analyzed these two heuristic algorithms for WSNs and proved self-stabilizing properties and step complexities that are $O(n^{2})$ and O(n) under the unfair scheduler.

We evaluated the performance of our algorithms together with their counterparts in the literature on randomly generated geometric networks and provided extensive performance analysis in different types of measurement including sent bytes, received bytes, energy consumption, move count and approximation ratio. The experimental results show that the SS-CVC${}_{2}$ and SS-CVC${}_{1}$ algorithms outperformed the existing matching based algorithms in terms of move counts, step counts and vertex cover solutions. On the other hand, the sent and received bytes of the SS-CVC${}_{1}$ algorithm are the lowest among all other algorithms. Through our extensive experiments, we can state that the SS-CVC${}_{1}$ algorithm is the most energy-efficient algorithm. The approximation ratio of the SS-CVC${}_{2}$ is not greater than 1.7 for all network types, while the other matching-based algorithms produced at least two times greater ratios than the optimal solution. In addition to these, we provided the performance results of the transformed version of SS-CVC${}_{2}$, which requires more time and energy to stabilize under an unfair scheduler but produces the same solution as SS-CVC${}_{2}$.

Conclusively, we can state that the SS-CVC${}_{2}$ algorithm is better than the others when 2-hop information is provided, in cases where the time and approximation ratio are the concern. However, if these are not provided, the SS-CVC${}_{1}$ algorithm is a very promising option to find the capacitated vertex cover rather than other matching-based vertex cover algorithms, accompanied by the advantage of energy efficiency. T-SS-CVC${}_{2}$ provides the same result in terms of vertex cover, but it needs more energy than SS-CVC${}_{2}$. In the future, we plan to study the hard capacitated version of the vertex cover and its application in real-world examples.

## Figures and Tables

**Figure 1 sensors-22-03774-f001:**
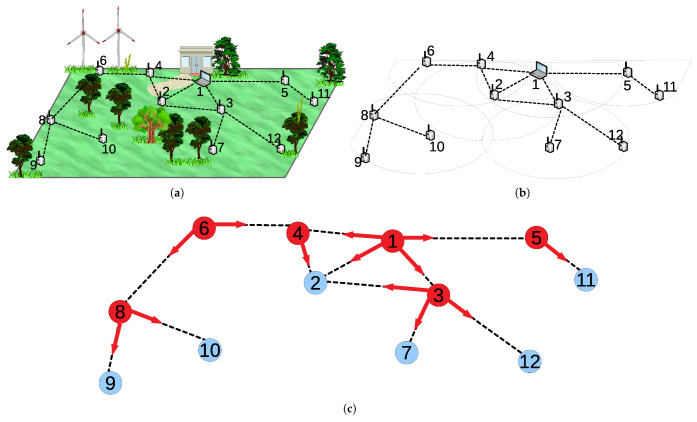
An example of link monitoring application for the vertex cover problem: (**a**) Deployment of a sample WSN; (**b**) Graph represantation of the topology; (**c**) Link monitoring application on the topology.

**Figure 2 sensors-22-03774-f002:**
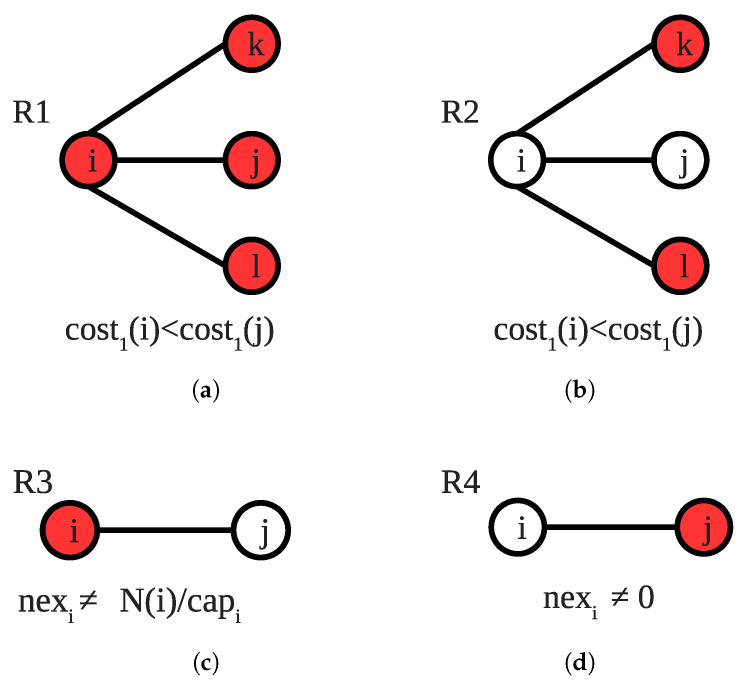
Sample scenarios for rules of SS-CVC${}_{1}$: (**a**) An Example for R1; (**b**) An Example for R2; (**c**) An Example for R3; (**d**) An Example for R4.

**Figure 3 sensors-22-03774-f003:**
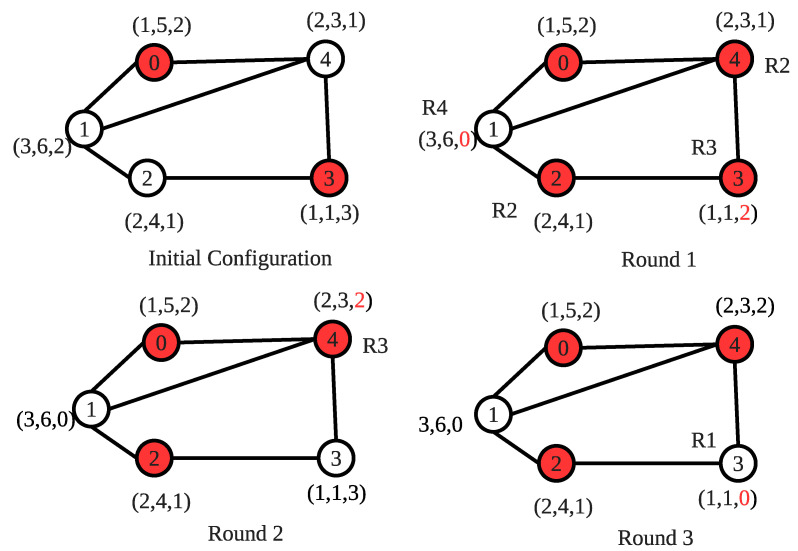
An execution of the SS-CVC${}_{1}$ algorithm.

**Figure 4 sensors-22-03774-f004:**
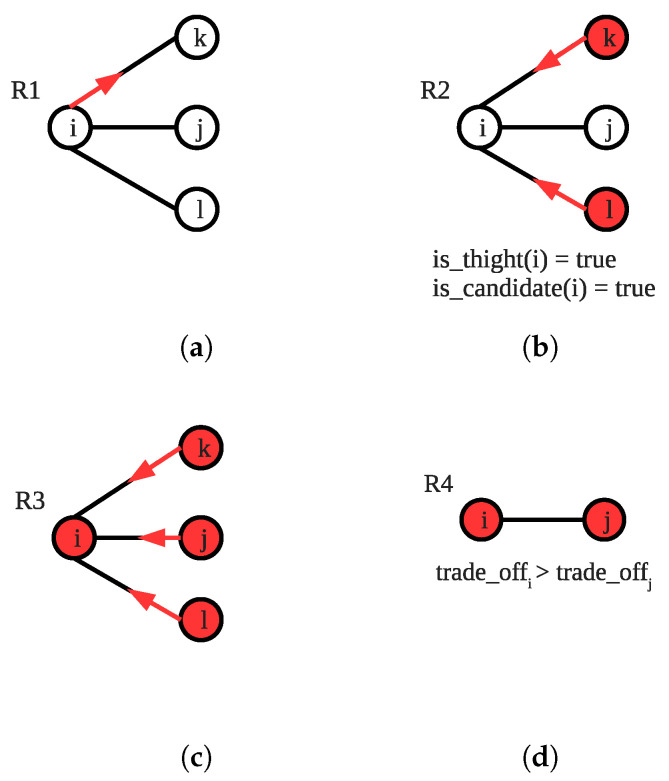
Sample scenarios for rules of SS-CVC${}_{2}$: (**a**) An Example for R1; (**b**) An Example for R2; (**c**) An Example for R3; (**d**) An Example for R4.

**Figure 5 sensors-22-03774-f005:**
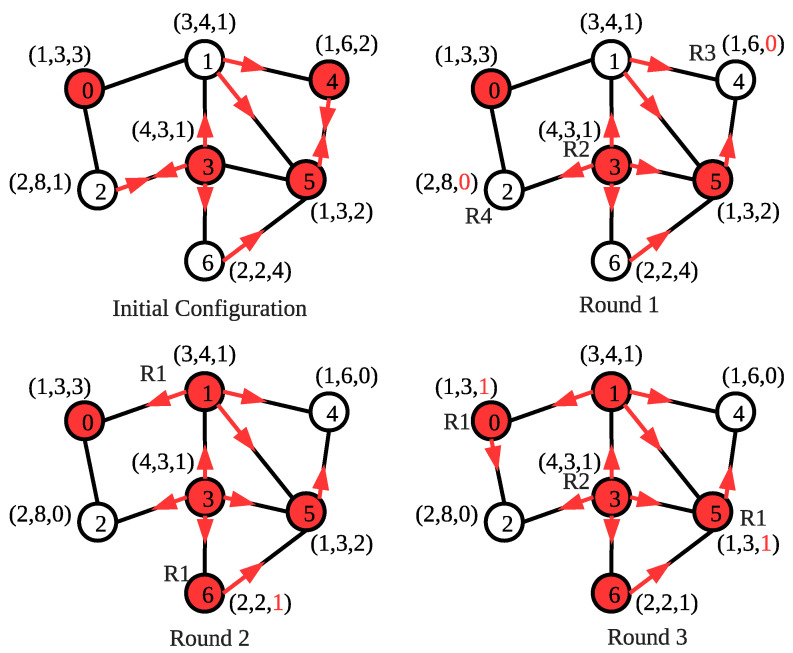
An execution of the SS-CVC${}_{2}$ algorithm.

**Figure 6 sensors-22-03774-f006:**
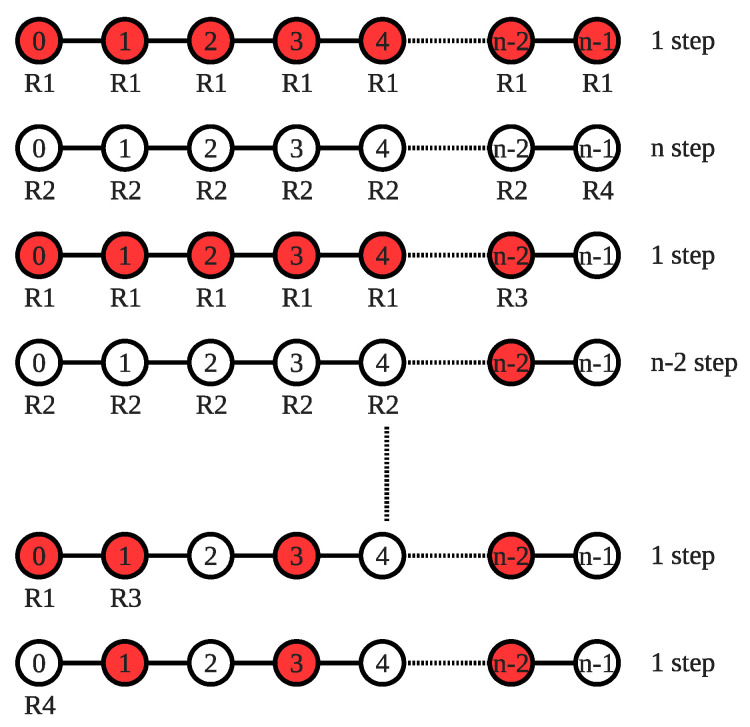
Worst-case scenario for the SS-CVC${}_{1}$ algorithm.

**Figure 7 sensors-22-03774-f007:**
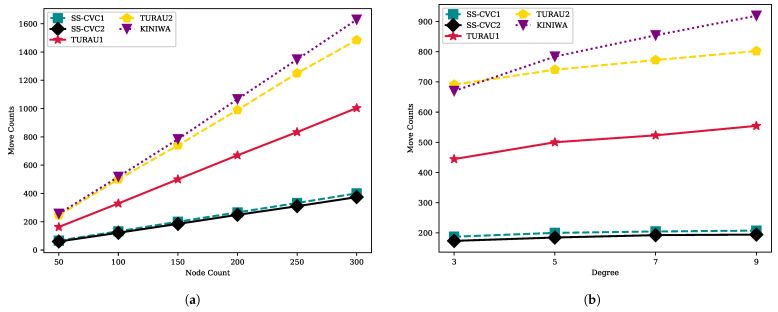
Move counts of algorithms on weighted networks: (**a**) Move count against node count on the fixed average degree 5; (**b**) Move count against node count on the fixed network size 150.

**Figure 8 sensors-22-03774-f008:**
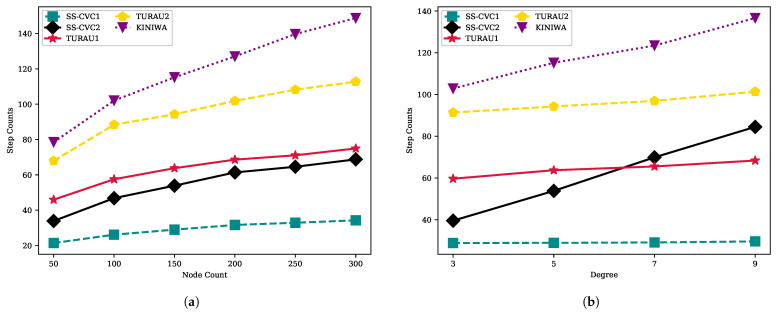
Move counts of algorithms on the weighted networks: (**a**) Step count against node count on the fixed average degree 5; (**b**) Step count against node count on the fixed network size 150.

**Figure 9 sensors-22-03774-f009:**
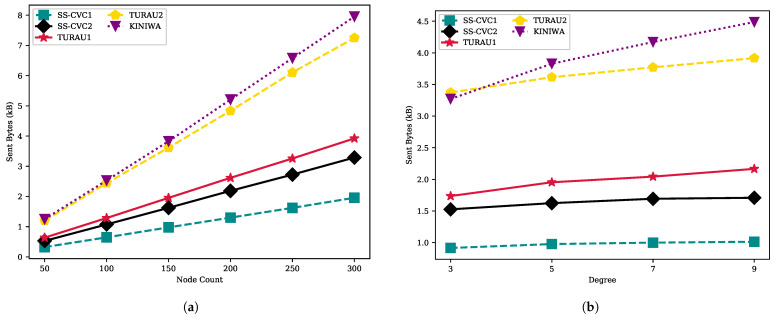
Sent bytes of algorithms on weighted networks: (**a**) Sent bytes against node count on the fixed average degree 5; (**b**) Sent bytes against node count on the fixed network size 150.

**Figure 10 sensors-22-03774-f010:**
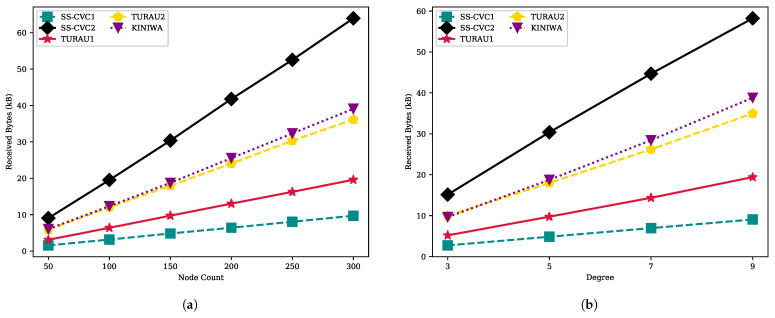
Received bytes of algorithms on weighted networks: (**a**) Received bytes against node count on the fixed average degree 5; (**b**) Received bytes against node count on the fixed network size 150.

**Figure 11 sensors-22-03774-f011:**
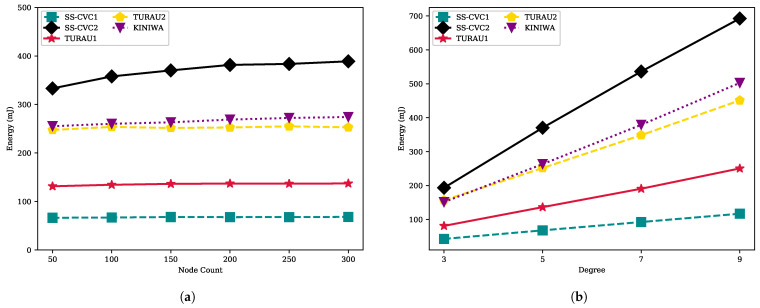
Energy consumption of algorithms on weighted networks: (**a**) Energy consumption against node count on the fixed average degree 5; (**b**) Energy consumption against node count on the fixed network size 150.

**Figure 12 sensors-22-03774-f012:**
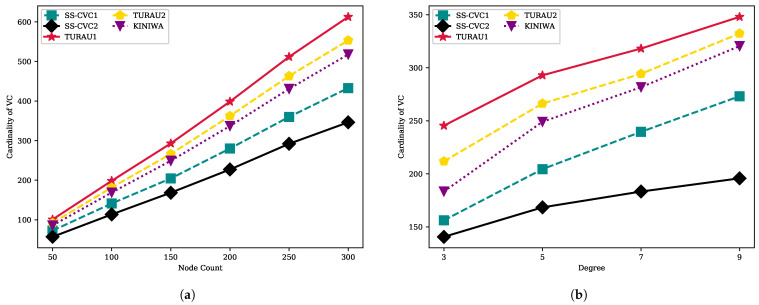
VC size of algorithms on the weighted networks: (**a**) VC size against node count on the fixed average degree 5; (**b**) VC size against node count on the fixed network size 150.

**Figure 13 sensors-22-03774-f013:**
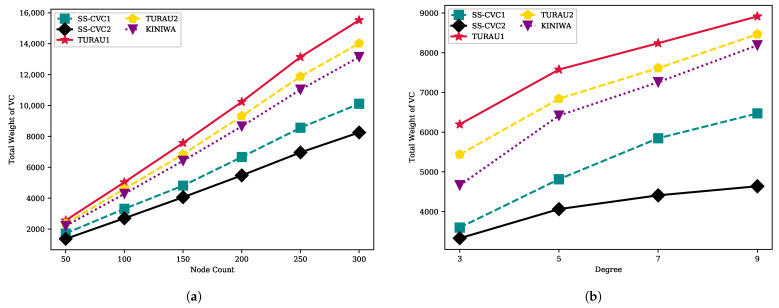
VC Weight of algorithms on the weighted networks: (**a**) VC weight against node count on the fixed average degree 5; (**b**) VC weight against node count on the fixed network size 150.

**Figure 14 sensors-22-03774-f014:**
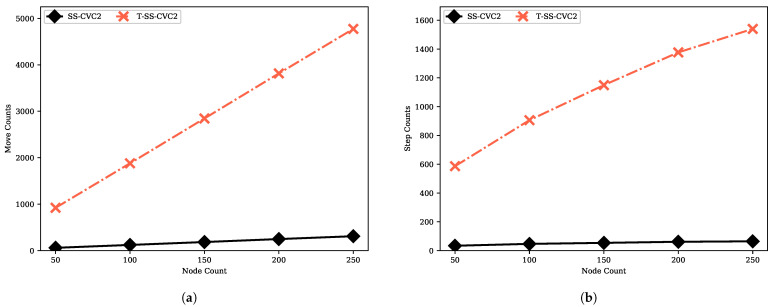
Move and step count performances of T-SS-CVC${}_{2}$ algorithm against SS-CVC${}_{2}$: (**a**) Move count against node count for the fixed average degree 5; (**b**) Step count against node count for the fixed average degree 5.

**Figure 15 sensors-22-03774-f015:**
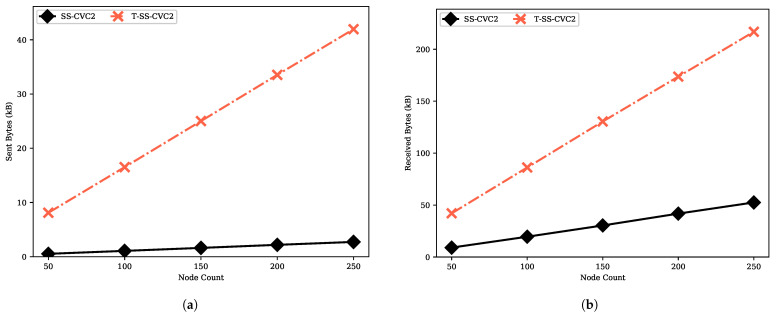
Sent and received bytes performances of the T-SS-CVC${}_{2}$ algorithm against SS-CVC${}_{2}$: (**a**) Sent bytes against node count for the fixed average degree 5; (**b**) Received bytes against node count for the fixed average degree 5.

**Figure 16 sensors-22-03774-f016:**
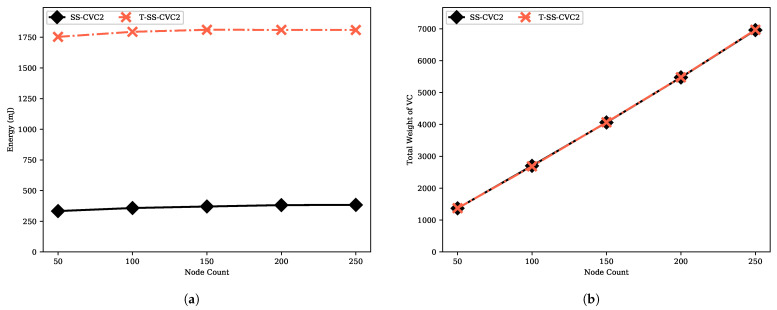
Energy consumption and solution performances of the T-SS-CVC${}_{2}$ algorithm against SS-CVC${}_{2}$: (**a**) Energy consumption against node count on the fixed average degree 5; (**b**) VC weight against node count on the fixed average degree 5.

**Table 1 sensors-22-03774-t001:** Approximation ratios of algorithms (Avg. degree = 3).

	SS-CVC${}_{2}$	SS-CVC${}_{1}$	KINIWA	TURAU${}_{2}$	TURAU${}_{1}$
50	1.55	1.71	2.17	2.50	2.88
100	1.53	1.65	2.16	2.48	2.90
150	1.56	1.69	2.19	2.56	2.92
200	1.56	1.71	2.18	2.54	2.92

**Table 2 sensors-22-03774-t002:** Approximation ratios of algorithms (Avg. degree = 5).

	SS-CVC${}_{2}$	SS-CVC${}_{1}$	KINIWA	TURAU${}_{2}$	TURAU${}_{1}$
50	1.62	2.03	2.61	2.77	3.06
100	1.61	1.98	2.56	2.75	3.03
150	1.61	1.93	2.54	2.70	2.99
200	1.62	1.96	2.55	2.74	3.03

**Table 3 sensors-22-03774-t003:** Approximation ratios of algorithms (Avg. degree = 7).

	SS-CVC${}_{2}$	SS-CVC${}_{1}$	KINIWA	TURAU${}_{2}$	TURAU${}_{1}$
50	1.67	2.23	2.78	2.99	3.18
100	1.66	2.18	2.74	2.84	3.07
150	1.63	2.16	2.68	2.83	3.03
200	1.62	2.11	2.67	2.78	3.01

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
