# Peer review of "Self-Stabilizing Capacitated Vertex Cover Algorithms for Internet-of-Things-Enabled Wireless Sensor Networks"

_sensors, 2022, doi:10.3390/s22103774_

Round 1

Reviewer 1 Report

The authors consider solving (?) (This is still unclear to me!) the capacitated vertex cover (VC) problem for wireless sensor networks (WSN). They propose two algorithms and show superiority over two other existing algorithms. This article might be very specialized, but definitely leaves many questions to the attentive reader. It is not made clear how the VC problem is exactly linked to WSN, how it comes up or why we need to solve it. Many notions such as self-stabilization, or the energy consumption are not properly defined, and it is still unclear to me why these are of importance or how they are measured. Last, the simulations show some results, but it is not clear if the two other algorithms that are included are the state of the art, or if the simulation setting is a recognized benchmark as opposed to some example data created by the authors, which would make all results much less meaningful. Importantly, the article seems to concentrate purely on the VC problem, and on solving it (optimally? not clear!), therefore how these results come into play again for WSN are not considered. Since the focus of the article is actually WSN, it is not sufficient to leave out WSN completely from the simulations, as all results in the simulations talk about VC solutions only.

abstract: How does fault tolerace help with battery drain? The motivation could be improved. The notion of "stabilized algorithm" has to be defined. The runtimes in O() notation are hard to interpret, is e.g. O(n) good or bad?
line 29: Is there a more concise definition of the notation of self-stabilization? The paragraph lists examples, but can it be more concrete?
line 41: "The scheduler decides which nodes could execute their moves according to two basic aspects such as spatially and temporarily." -- This sentence is incomplete.
line 51: It is not clear why vertex cover (VC) is important: Why does it come into play for routing or clustering, and what is the VC result used for or how is it interpreted?
line 123: It is not clear to me what exactly the self-stabilizing property is for VC that the authors refer to. Surely the notion of self-stabilization has been mentioned, but how exactly is a VC algorithm stabilized, aka under which external pertubations? Overall, I feel it is not entirely clear still what *exactly* the aim is here.
line 141: The VC problem has already been introduced.
line 158: What is a 1-hop neighbor? Do the authors mean a degree-1 neighbor?
line 162: All these notions are not introduced properly, and that makes the text hard to understand. However, the reader should be guided clearly through the manuscript, without guessing. The authors have defined the vertex cover (VC) problem on a graph, that was all good. Now, how is there the energy consumption coming in? And the energy of what? Of a machine running a WSN, of a machine solving the VC problem, and on what graph? All this is not clear. Likewise the time usage of what, of running WSN, of solving VC and to what accuracy (exactly or approximately)? Again all unclear. Regarding the SS-CVC algorithm, how is a stable configuration defined? Is it just any solution? Moreover, "the SS-CVC algorithms should be energy efficient" does not make any sense at all. How is an algorithm energy efficient? An algorithm is given in pseudo-code and implemented on some platform. Whether or not it is energy efficient could potentially be defined once we know the architecture it runs on, but the authors fail to mention any of this crucial information. Likewise, in "When a failure occurs, each node..." it is entirely unclear what a failure is! How is all this defined? Is it a bit-flip in the WSN, a wrong result or transmission, etc?
line 187: Algorithm1 is wrongly specified. An algorithm has steps 1 to X and says what to do in each step. In Algorithm1, it is not clear what to do with the rules! Do you apply each once? Are they mutually exclusive? Do you repeat them until no rule applies any more? All this again has to be specified. Even worse, it is totally unclear what Algorithm1 is even supposed to do? Is it a heuristic for solving the VC problem?
line 227: It seems to me that both Algorithm1 and Algorithm2 are heuristics for the VC problem. Can this be clearly stated in the manuscript?
line 317: Is the claim here that the stable configuration of Algorithm2 is an optimal solution of the VC problem?
line 351: It might be good to also briefly explain here how the competitior algorithms (Kiniwa and Tureau) work.
line 418: It does not seem clear to me how the authors measure the sent and received bytes for the messages. Can this be highlighted?
line 435: Can the authors motivate why eq. (4) reflects somehow the energy consumption? Where do all the seemingly random coefficients come from, and why is the value of eq. (4) important?
line 498: The authors claim that their proposed algorithms outperform the existing ones, but here any reader would have more questions: Can the authors motivate that Kiniwa and Tureau are the state of the art? And the data/ simulation setting they considered in their study, is that a recognized benchmark for the problem, or some example the authors created and on which their method works better? In summary, claiming superiority based on one example dataset is not convincing, the study has to be much more extensive or a recognized benchmark in the literature has to be used.

Author Response

Dear Respected Reviewer,

Best Regards,

Reviewer 2 Report

My Comments and Suggestions to Authors:

  1. The article needs to be revised with more experimentation by comparing relevant approaches and algorithms.

2. An introduction should clearly highlight the motivation, problem statement, the objective of the paper, gap in the existing research and the novelty of the conducted research.  

3. When I checked the results, I noticed that there were mistakes, please recheck.  

4. I suggest extending the conclusions section to focus on the results you get, the method you propose, and their significance.  

5.   Expand literature review by citing related work:   Certain algorithms for computing strength of competition in bipolar fuzzy graphs, International Journal of Uncertainty, Fuzziness and Knowledge-Based Systems, 25(6)(2017), 877-896  An optimization study based on Dijkstra algorithm for a network with trapezoidal picture fuzzy numbers. Neural Comput & Applic 33, 1329–1342 (2021)  

Author Response

Dear Respected Reviewer,

Best Regards,

Round 2

Reviewer 1 Report

The reviewer would like to thank the authors for their reply and for addressing many comments of the first review round. However, upon re-reading the manuscript, I'd have the following comments:

Section1
===============
"Therefore, the use of fault tolerance techniques such as self-stabilization @R1C4 that brings the network back to its legitimate state when the topology is changed due to node leaves."
-- The use of fault tolerance techniques is what?

"A crucial theoretical graph structure is the vertex cover that can be used in various WSN applications such as routing, clustering, replica placement, and link monitoring."
-- Why is it crucial?

"Capacitated vertex cover is the generalized version of the problem which restricts the number of edges covered by a vertex."
-- Restricts in which way?

"In this paper, we propose two self-stabilizing algorithms for the capacitated vertex cover problem."
-- You propose two algorithms for what? To solve capacitated vertex cover? It is not true that those are the first algorithms to tackle this problem! See
https://www.sciencedirect.com/science/article/abs/pii/S0196677403000531

"The first algorithm is stabilized under an unfair distributed scheduler at most O(n2 ) step where n is the count of nodes. The second algorithm assumes 2-hop knowledge about the network and runs under the unfair scheduler, which subsumes the synchronous and distributed fair scheduler and stabilizes itself after O(n) moves in O(n) step"
-- Where does a scheduler come from? That has not even be mentioned beforehand? What is 2-hop knowledge, and why is it coming up here? How do you define an unfair scheduler? What is the stabilization of a scheduler? So many items that come from nowhere and are not introduced!

"which means the time complexity of the algorithm increases linearly (that is acceptable for most WSN setups) by the size of the graph"
-- This is trivial, you introduced n as the number of vertices before.

"The gathered measurements revealed that the proposed algorithms are faster than their competitors, use less energy and offer better vertex cover solutions."
-- Use less energy? Where does energy come from? Algorithms are analyzed theoretical with O() complexity. Why do you consider energy consumption and how is it measured?

"In such environments, WSNs are vulnerable in the manner of the communication of the nodes and the sustainability of the network."
-- What does this mean?

"A vertex cover of a given undirected graph G(V,E) is a set VC"
-- The definition that VC is a subset of V should go here, not at the end of the sentence.

"The optimization version of the vertex cover problem is in the NP-Hard complexity class"
-- Optimized for what? The maximal vertex cover is simply the whole vertex set V and that is trivial to solve. It only becomes NP when you want a minimal vertex cover, but that is all not described here, and it is unclear why the authors even consider the minimal vertex cover problem.

"Although there are notable number of VC related studies, capacitated version of the problem is merely investigated in the literature."
-- What does "merely" mean? It is investigated too in the literature, thus it would be crucial if the authors say why they solve it again and don't use one of the existing algorithms for the capacitated vertex cover.

Section2
===============
"The VC problem cannot be approximated within a ratio of 1.36"
-- Is that for the minimal vertex cover?

Literature review is well made! Thank you!
However, it only takes about vertex cover! Where is the WSN?

"Gandhi et al. have introduced a 2-approximation algorithm which consists of a pre-processing step to reduce the vertex cover size."
-- A reference is missing.

"each capacity-1 vertex is excluded from the solution set is executed"
-- That sentence does not make sense.

Section 3
===============
Fig1 is good!

"In this study, our objective is to develop self-stabilizing capacitated vertex cover algorithms (SS-CVC) that are efficient in terms of energy consumption, time usage, and the size of the vertex cover set."
-- Again, either before or right after this sentence (which isn't the case!) one needs to introduce all the notions being used. Efficiency is defined how? What is even energy consumption? Of the computers being used, or the sensors etc? What is time usage of a sensor, isn't that thing on all day? What is the size of the vertex cover set? You NEVER said that you consider the minimal vertex cover problem, which is the NP version! You have to be much more precise here!

"To fulfill the convergence property, the algorithms must reach the stable configuration (in which nodes do not make a move) as fast as they can within a minimum number of moves and steps."
-- This is wrong, and mathematically not well defined. What is a stable configuration and can you define it mathematically, please? That has all to be introduced! Convergence does not mean in a minimal number of steps, you are implicitly introducing another optimization problem here! How is it well-defined? What is the objective function that you try to optimize/solve?

Section4
===============
Section4.1: The authors still not say if they consider an exact solution or a heuristic/ approximation solution!

Also they only solve vertex cover! What is the connection to WSN? Don't you want to actually solve that? Both do not seem clear to me at all.

It is not clear what the output of the two algorithms is: Is it a binary indicator per vertex of whether that vertex belongs to the vertex cover?

"Each vertex contributes to the solution set only once; therefore, the total weight of this solution is 15, that is, the optimal solution."
-- Is optimality a general statement here?

Section5
===============
According to Thm 5.5, the solution is a minimal capacitated vertex cover.
-- That is the first time I am reading that the authors consider the minimal vertex cover! That is in NP.

In Thm 5.6 they claim O(n) time complexity. So have the authors solved P=NP now that they presented a minimal vertex cover algorithm in O(n) ?

Section6
===============
How do WSNs now come into play after considering only vertex cover beforehand?

In Section 6.1 the authors state " The energy consumption of an algorithm is the energy required for its execution."
-- This would be needed way earlier, aka in the intro. Also, how did they measure it, by tracking the energy consumption of their laptop? How do you define the energy consumption of the WSN then, with all sensors and more pieces?

Are the linear relationships in e.g. Fig10 expected?

How is eq. (4) derived? Where do the constants come from? Where they measured somehow on the device that the authors used?
If yes to the latter, then how transferable are the results to other users and/or systems?

Section7
===============
"The experimental results show that SS-CVC2 and SS-CVC1 algorithms overperformed"
-- Do you mean "outperformed"?

"The approximation ratio of SS-CVC2 is not greater than 1.7 for all network types, while the other matching-based algorithms produced at least 2 times greater than the optimal solution."
-- How was the optimal solution determined and known in these cases?

Author Response

Dear Respected Reviewer,

The responses are attached.

Best Regards,

Reviewer 2 Report

This version is better. 

Author Response

Dear Respected Reviewer,

Thanks for your decision and your suggestions that greatly improved the quality of the paper.

Best Regards,

Round 3

Reviewer 1 Report

The reviewer would like to thank the authors for their point-by-point responses and their improvements of the manuscript.